# Structure of CFTR bound to (*R*)-BPO-27 unveils a pore-blockage mechanism

Paul G. Young [1,2], Karol Fiedorczuk[1] & Jue Chen [1,3] ✉

Hyperactivation of the cystic fibrosis transmembrane conductance regulator (CFTR) contributes to secretory diarrhea, a major cause of pediatric mortality worldwide, and autosomal dominant polycystic kidney disease (ADPKD), the most common inherited cause of end-stage renal disease. Selective CFTR inhibition is a potential therapeutic strategy, with (*R*)-BPO-27 emerging as a promising candidate. Here, we present a cryo-EM structure of CFTR bound to (*R*)-BPO-27 at an overall resolution of 2.1 Å. Contrary to the previous hypothesis that it inhibits CFTR current by competition with ATP, we demonstrate that (*R*)-BPO-27 instead directly occludes the chloride-conducting pore while permitting ATP hydrolysis, thus uncoupling the two activities. Furthermore, we find that inhibitor binding requires some degree of NBD separation, as the inhibition rate inversely correlates with the probability NBD dimerization. These findings clarify the compound's mechanism and provide a molecular basis for optimizing its clinical potential.

The cystic fibrosis transmembrane conductance regulator (CFTR) is an ATP-gated anion channel that plays a major role in salt and fluid homeostasis throughout epithelial tissues[1]. CFTR is activated by phosphorylation via cyclic nucleotide-dependent protein kinases, and channel opening allows conductance of chloride down its electrochemical gradient[2–6]. Whereas defects in CFTR function cause the inherited disease cystic fibrosis (CF)[7], its hyperactivation leads to aberrant fluid secretion into the gut lumen in secretory diarrheas and into renal cyst lumina in autosomal dominant polycystic kidney disease (ADPKD)[8–15]. Thus, CFTR is an important therapeutic target for multiple diseases.

The structure and function of CFTR are well characterized[16]. Despite functioning as a channel, CFTR structurally resembles ATP-driven transporters in the ATP-binding cassette (ABC) transporter superfamily[17]. It consists of two nucleotide-binding domains (NBDs), two transmembrane domains (TMDs), and a unique regulatory (R) domain[18]. The dephosphorylated R domain functions as an auto-inhibitory element by preventing NBD dimerization[2]. Upon phosphorylation by protein kinases, the R domain is displaced, permitting ATP-dependent conformational changes necessary to gate the pore[19,20].

Small molecules that enhance CFTR folding (correctors) or activity (potentiators) have advanced to the clinic for cystic fibrosis treatment[21–28]. In contrast, CFTR inhibitors remain in early development despite their therapeutic potential for secretory diarrheas and ADPKD[29,30]. Several small-molecule inhibitors have been identified, including CFTR$_{inh}$-172 and (*R*)-BPO-27[31–36]. Although CFTR$_{inh}$-172 is well-characterized and has been a valuable research tool[37–39], its low solubility and lack of specificity have hindered its clinical use[40–42]. In contrast, (*R*)-BPO-27 is an attractive clinical candidate for its low-nanomolar potency, high bioavailability, low toxicity, and high water solubility[36,43]. In animal models, (*R*)-BPO-27 has shown efficacy in preventing *Escherichia coli*- and *Vibrio cholerae*-mediated secretory diarrheas[44].

However, the mechanism by which (*R*)-BPO-27 inhibits CFTR remains poorly understood. Its binding site has not been identified, and its structural impact on the channel is uncharacterized. Computational modeling and electrophysiology measurements suggest that it competes with ATP binding at the NBDs[45], but direct experimental evidence supporting this mechanism is lacking. Defining how (*R*)-BPO-27 interacts with CFTR at the molecular level is crucial for elucidating its mechanism and guiding therapeutic optimization.

[1]Laboratory of Membrane Biology and Biophysics, The Rockefeller University, New York, NY, USA. [2]Weill Cornell/Rockefeller/Sloan Kettering Tri-Institutional MD-PhD Program, New York, NY, USA. [3]Howard Hughes Medical Institute, The Rockefeller University, New York, NY, USA. ✉e-mail: juechen@rockefeller.edu

In this study, we use cryogenic electron microscopy (cryo-EM) and electrophysiology to identify the binding site and mechanism of (*R*)-BPO-27. Contrary to the model suggesting ATP-competitive inhibition, our structural analysis reveals that (*R*)-BPO-27 binds within the pore and sterically blocks ion conductance. Despite its binding, the maximal rate of ATP hydrolysis remains unaffected. Thus, (*R*)-BPO-27 uncouples channel gating from enzymatic activity. These findings advance our understanding of (*R*)-BPO-27 inhibition and offer a molecular framework to analyze its structure-activity relationship.

## Results

### (*R*)-BPO-27 uncouples CFTR channel activity from ATP hydrolysis

In the CFTR gating cycle, ATP hydrolysis and channel gating are tightly coupled. Once phosphorylated, ATP binding opens the chloride-conducting pore, while subsequent ATP hydrolysis leads to its closure[16,46]. Amino acid substitutions that attenuate ATP binding reduce channel activity, while those that impair ATP hydrolysis prolong channel opening[47]. Although (*R*)-BPO-27 is a well-established inhibitor of CFTR channel activity[36,45,48], its effect on ATP hydrolysis remains uncharacterized.

As (*R*)-BPO-27 was proposed to inhibit CFTR current by competing with ATP for binding[45], one would expect it to inhibit ATP hydrolysis as well. To test this, we conducted parallel electrophysiological and ATP hydrolysis measurements to directly compare the effects of (*R*)-BPO-27 on the two functional activities of CFTR. In both experiments, we used 1 μM (*R*)-BPO-27, which is a saturating concentration in electrophysiology and 2000-fold higher than the reported $IC_{50}$[45]. In inside-out membrane patches, (*R*)-BPO-27 abolished wild-type (WT) CFTR current with an inhibition half-time ($t_{1/2,\ inh}$) of roughly 12 s. This inhibition was reversible with extensive washing, although the inhibitor exhibited a very slow apparent off-rate ($t_{1/2,\ rec}$ ~ 420 s) (Fig. 1A).

In steady-state ATP hydrolysis measurements, 1 μM (*R*)-BPO-27 increased the $K_m$ value of ATP from 0.36 ± 0.11 mM to 1.6 ± 0.4 mM (Fig. 1B). This shift in $K_m$ aligns well with previous electrophysiology studies[45], indicating that the effect of (*R*)-BPO-27 on CFTR is comparable in both detergent-solubilized and membrane environments.

However, the data are inconsistent with ATP competition representing the primary mechanism of inhibition, as 1 μM (*R*)-BPO-27 is sufficient to fully inhibit CFTR current at a saturating ATP concentration (3 mM) (Fig. 1A) but has no effect on the rate of ATP hydrolysis at the same ATP concentration (Fig. 1B, C). This contrasts with CFTR$_{inh}$-172, which inhibits ATP hydrolysis by 72% and current by 95% at a saturating inhibitor concentration (10 μM) (Fig. 1C, D)[37–39]. Furthermore, if (*R*)-BPO-27 were competing with ATP, one would expect the apparent $K_m$ for ATP to increase proportionally with inhibitor concentration[49]. However, despite a tenfold change in inhibitor concentration from 0.5 μM to 5 μM, the apparent $K_m$ changed only modestly from 1.4 ± 0.6 mM for 0.5 μM (*R*)-BPO-27 to 1.8 ± 0.8 mM for 5 μM. These observations suggest that (*R*)-BPO-27 does not act through a direct competitive mechanism.

### The structure of the CFTR/(*R*)-BPO-27 complex reveals a binding site inside the pore

To identify the binding site of (*R*)-BPO-27, we determined the cryo-EM structure of inhibitor-bound CFTR using a hydrolytically inactive CFTR variant (E1371Q) (Fig. S1; Table S1). In the presence of saturating concentrations of ATP (8 mM) and (*R*)-BPO-27 (92 μM), CFTR (E1371Q) adopts an NBD-dimerized conformation similar to previously reported structures of CFTR (E1371Q) obtained without inhibitor (Fig. 2A; Fig. S2; RMSD 0.825 Å). The structure was determined to an overall resolution of 2.1 Å, thus densities for side chains and ligands are clearly defined (Fig. S1; Table S1). Within the CFTR pore, we observed a prominent non-protein density with clear features corresponding to (*R*)-BPO-27 (Fig. 2A, B). Consistent with previous studies showing that (*R*)-BPO-27, but not its (*S*) enantiomer, inhibits CFTR[48], modeling (*S*)-BPO-27 into the density resulted in the 5-bromofuran and carboxylate groups projecting outside the experimental density (Fig. 2C).

The molecular details of how (*R*)-BPO-27 interacts with CFTR enable us to better understand its structure-activity relationship (SAR). (*R*)-BPO-27 binds within the inner vestibule of the CFTR pore at the level of the membrane's outer leaflet, forming extensive interactions with residues from six transmembrane (TM) helices (Fig. 3A–C). Its

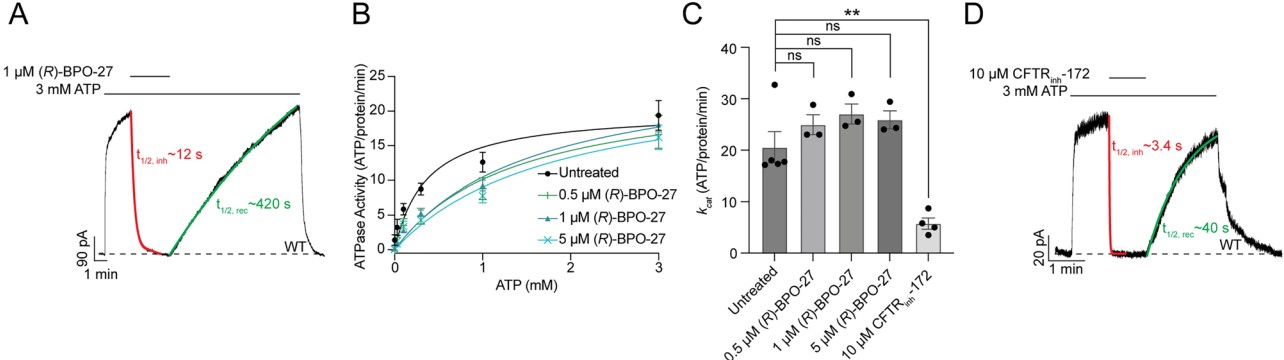

**Fig. 1 | (*R*)-BPO-27 uncouples chloride conductance and ATP hydrolysis.**
**A** Representative macroscopic current recording from an inside-out membrane patch excised from cells expressing WT CFTR. Patches exposed to 1 μM (*R*)-BPO-27 demonstrated a $t_{1/2}$ of 12 ± 0.6 s for inhibition and a $t_{1/2}$ of 420 ± 75 s for recovery. In (**A**) and (**D**), mean $t_{1/2}$ values were calculated by fitting one-phase exponential decay functions for inhibition and one-phase exponential association functions for recovery to each replicate and averaging the resulting $t_{1/2}$ values. Data represent the mean and standard error (SE) from *n* = 5 measurements. **B** Effect of (*R*)-BPO-27 on steady-state ATP hydrolysis by PKA-phosphorylated WT CFTR. Data represent means and SE from *n* = 3 (0.5 μM, 1 μM, and 5 μM (*R*)-BPO-27) or *n* = 5 (untreated) measurements and are fitted with the Michaelis–Menten equation. Without (*R*)-BPO-27, the $K_m$ was 0.36 ± 0.11 mM; with 0.5 μM (*R*)-BPO-27, the $K_m$ was 1.4 ± 0.6 mM; with 1 μM (*R*)-BPO-27, the $K_m$ was 1.6 ± 0.4 mM; with 5 μM (*R*)-BPO-27, the $K_m$ was 1.8 ± 0.8 mM. The $k_{cat}$ values are plotted in (**C**). **C** Maximal turnover rates

of WT CFTR in the absence and presence of inhibitors. The $k_{cat}$ values based on fitted Michaelis–Menten curves: in the absence of inhibitors: 21 ± 3 ATP/protein/minute; in the presence of 0.5 μM (*R*)-BPO-27: 25 ± 2 ATP/protein/minute; in the presence of 1 μM (*R*)-BPO-27: 27 ± 2 ATP/protein/minute; in the presence of 5 μM (*R*)-BPO-27: 26 ± 2 ATP/protein/minute; in the presence of 10 μM CFTR$_{inh}$-172: 5.8 ± 1.1 ATP/protein/minute. Data represent means and SE from *n* = 3 (with all concentrations of (*R*)-BPO-27), *n* = 4 (with CFTR$_{inh}$-172), or *n* = 5 (untreated) titrations. Statistical significance was tested by one-way ANOVA with correction for multiple comparisons (ns: not significant; **$P = 10^{-3}$). **D** Representative macroscopic current recording from an inside-out membrane patch excised from cells expressing WT CFTR exposed to 10 μM CFTR$_{inh}$-172. Patches exposed to 10 μM CFTR$_{inh}$-172 demonstrated a $t_{1/2}$ of 3.4 ± 0.9 s for inhibition and a $t_{1/2}$ of 40 ± 11 s for recovery. Data represent means and SE from *n* = 6 measurements for inhibition and *n* = 5 measurements for recovery.

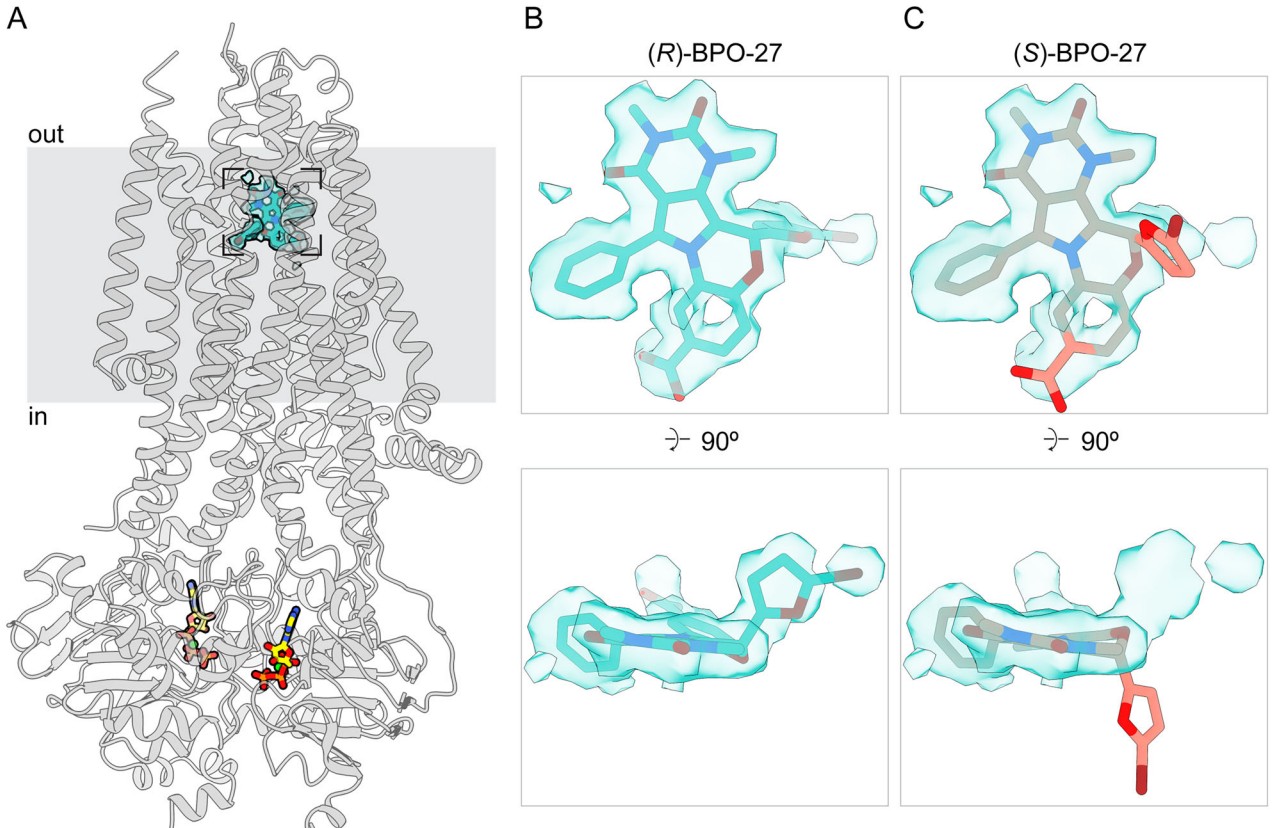

**Fig. 2 | The structure of CFTR/(R)-BPO-27 complex. A** The overall structure. (R)-BPO-27 is shown in teal surrounded by cryo-EM density. ATP is shown as yellow stick models. **B** Stick model of (R)-BPO-27 surrounded by cryo-EM density. Two views are shown. **C** The (S)-enantiomer does not fit into the cryo-EM density. The protruding 5-bromofuran and carboxylate groups are shown in red. Two views are shown.

central benzopyrimido-pyrrolo-oxazinedione group occupies a large polar cavity formed by TMs 1, 6, and 12. The carboxylate moiety at the 7' position on the benzopyrimido ring forms a salt bridge with K95 (Fig. 3A). Prior SAR studies have shown that although compounds with nonpolar groups at the 7' position can inhibit CFTR, the highest affinity is achieved when the group is a polar substituent, such as $-NO_2$ or $-COOH$[36]. We also found that substitution of K95 with alanine significantly reduced the inhibitory efficacy of $1\,\mu M$ (R)-BPO-27 (Fig. 3D, E). These observations indicate that this salt bridge strongly contributes to the binding energy of the compound.

Extending beyond the central cavity, (R)-BPO-27 interacts with two hydrophobic pockets through its phenyl and 5-bromofuran groups (Fig. 3A–C). On one side, the phenyl group fits into a pocket formed by P99 from TM1 and I344 and V345 from TM6 (Fig. 3A, C). Although this pocket is shallow, (R)-BPO-27 analogs lacking the phenyl group had no activity[35], underscoring the importance of these interactions. On the other side of the benzopyrimido-pyrrolo-oxazinedione group, the 5-bromofuran moiety fits into a pocket formed by residues from TMs 5, 10, and 12 (Fig. 3B, C). SAR studies revealed that substituting the Br atom at the 5' position of the furan ring with bulkier groups, such as phenyl, primary alcohol, or propyl, resulted in inactive compounds[36]. The structure shows that the 5-bromofuran group fits snugly into a pocket formed by F310, F311, L997, I1000, and W1145, explaining why larger groups at this position would be incompatible with binding (Fig. 3B, C). The inactivity of (S)-BPO-27 further demonstrates the importance of these interactions, as modeling this enantiomer into the binding site flips the 5-bromofuran group completely out of the hydrophobic pocket, thus eliminating these contacts and causing a significant clash with TM 12 (Fig. 3F)[48].

## (R)-BPO-27 inhibits CFTR by occluding the pore

Binding of (R)-BPO-27 did not induce any conformational change in CFTR but instead resulted in a complete blockage of the pore (Fig. S2; Fig. 4A). In the uninhibited, open-channel state, dehydrated chloride ions can traverse the lipid bilayer via a continuous pathway that includes a large inner vestibule, a selectivity filter, and a narrow extracellular portal (Fig. 4A)[50]. Binding of (R)-BPO-27 inside the inner vestibule sterically interrupts ion flow through this pathway (Fig. 4A). This blockage is particularly effective as the inhibitor occupies a key region within the vestibule at the entrance to the selectivity filter[50]. As a result, ion flux is entirely inhibited despite the channel remaining in a conformation nearly identical to that of the open-channel structure.

In agreement with our biochemical data suggesting that (R)-BPO-27 does not compete with ATP, we observed no density corresponding to the inhibitor in the NBDs. Instead, the high-resolution map clearly shows that ATP-$Mg^{2+}$ and water molecules are bound at both nucleotide-binding sites (Fig. 4B). Furthermore, the structure of the NBD dimer closely resembles the canonical structure, with no observable alterations at the ATP-binding sites (Fig. S2). These findings strongly suggest that (R)-BPO-27 inhibits CFTR by physically obstructing the ion conduction pathway, rather than by competing with ATP.

## (R)-BPO-27 association rate is inversely correlated with the probability of NBD dimerization

Given that (R)-BPO-27 is larger than the intracellular opening of the pore in the NBD-dimerized conformation, we hypothesized that its binding requires some degree of NBD separation. This hypothesis predicts that the kinetics of inhibition should be inversely related to the probability of NBD dimerization. To test this prediction, we

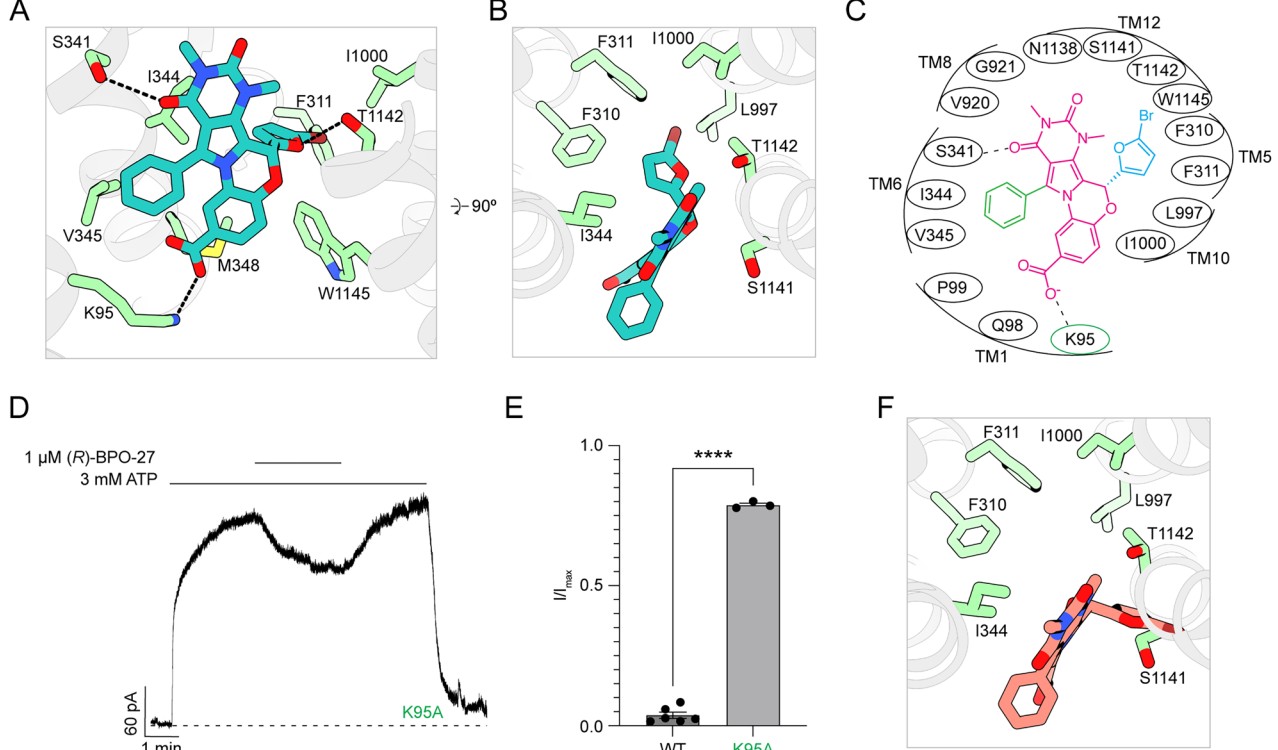

**Fig. 3 | (R)-BPO-27 binds within the inner vestibule. A** The binding site of (R)-BPO-27 with the inhibitor shown as teal sticks. Side chains within 4.5 Å of the inhibitor are shown in green. Hydrogen bonds are shown as black dashed lines. **B** View of the (R)-BPO-27 binding site looking down the long axis of the pore from the extracellular side. **C** Schematic drawing of CFTR-inhibitor interactions. All residues within 4.5 Å of the inhibitor are shown. The benzopyrimido-pyrrolo-oxazinedione group is shown in magenta, the phenyl group is shown in green, and the 5-bromofuran group is shown in blue. K95A is indicated by a green circle. **D** Representative macroscopic current recording from an inside-out membrane patch excised from cells expressing CFTR (K95A). **E** Inhibition of WT CFTR and CFTR (K95A) by 1 μM (R)-BPO-27. Data represent mean normalized current and SE from n = 6 (WT) or n = 3 (K95A) patches. The current following addition of 1 μM (R)-BPO-27 was normalized to the current prior to inhibitor addition. Statistical significance was tested using a two-tailed Student's t-test (****$P < 10^{-4}$). **F** The (S) enantiomer of (R)-BPO-27 modeled into the (R)-BPO-27 density viewed down the long axis of the pore from the extracellular end. The (S) enantiomer is shown as red sticks.

compared the rates of inhibition by 10 μM (R)-BPO-27 across three CFTR variants with distinct NBD dimerization probabilities. In the presence of 3 mM ATP, CFTR (E1371Q) remains constitutively NBD-dimerized, WT CFTR has an NBD dimerization probability of 85%, and CFTR (W401A) exhibits a probability of 50%[19].

Consistent with our hypothesis, 10 μM (R)-BPO-27 inhibits CFTR (W401A) approximately twofold faster than WT CFTR ($t_{1/2, WT} = 8.3 \pm 0.7$ s; $t_{1/2, W401A} = 4.5 \pm 0.8$ s; Fig. 4C, D). In contrast, the inhibition rate for CFTR (E1371Q) is much slower ($t_{1/2, E1371Q} = 230 \pm 18$ s), and inhibition occurs only upon ATP withdrawal (Fig. 4C, D). Since CFTR (E1371Q) stabilizes the NBD dimer and ATP withdrawal is necessary for NBD dissociation[50,51], these observations suggest that (R)-BPO-27 binding requires CFTR to transition away from a fully NBD-dimerized state.

Once inhibited, both WT CFTR and CFTR (W401A) recover current slowly following (R)-BPO-27 washout in the presence of ATP, with $t_{1/2}$ values of $340 \pm 51$ s and $330 \pm 53$ s, respectively (Fig. S3A, B). Furthermore, CFTR (E1371Q) recovery requires simultaneous withdrawal of ATP and the inhibitor, followed by re-exposure to ATP (Fig. 4C; Fig. S3C). Thus, for all three variants the rate of current recovery from (R)-BPO-27 inhibition is orders of magnitude slower than that of NBD separation in the absence of inhibitor[19]. Taken together, these findings suggest that (R)-BPO-27 binding stabilizes the NBD dimer and traps the channel in an inhibited state.

## Discussion

In this study, we provide structural evidence that (R)-BPO-27 functions as a pore blocker of the CFTR channel. It binds within a large pocket near the narrow region of the pore, sterically obstructing ion access to the selectivity filter and thereby inhibiting chloride conduction. The binding site aligns well with previous SAR data, emphasizing the critical roles of a salt bridge and two hydrophobic pockets in ligand recognition.

Given the large size of (R)-BPO-27 relative to the cytoplasmic pore opening (Fig. 4), its access to the binding site likely requires separation of the NBDs to open the cytosolic route. The inverse correlation between inhibition rate and the probability of NBD dimerization supports this notion. However, mapping the inhibitor-binding site onto the dephosphorylated, NBD-separated CFTR structure reveals that TM helices 8 and 12, which undergo local rearrangements upon NBD dimerization[52,53], would obstruct the (R)-BPO-27 binding site in the NBD-separated state (Fig. S4). These observations suggest that (R)-BPO-27 binds to an intermediate sampled during the transition from the NBD-separated state to the fully NBD-dimerized state stabilized by the E1371Q substitution. Elucidating the molecular details of how the inhibitor engages this intermediate remains an active focus of our ongoing research.

While the structure shows that (R)-BPO-27 binds deep in the pore, distant from the ATP-binding sites, functional assays reveal modest right-shifts in both the $EC_{50}$ for ATP-gated current[45] and the apparent $K_m$ for ATP hydrolysis (Fig. 1B). Although a change in $K_m$ is often evidence for competitive inhibition, two observations contradict that interpretation here. First, if (R)-BPO-27 were a true competitive inhibitor, its $IC_{50}$ of 0.5 nM, converted to a $K_i$ with the Cheng-Prusoff equation[54], predicts that 1 μM inhibitor should push the apparent $K_m$ for ATP into the molar range, which is orders of magnitude greater

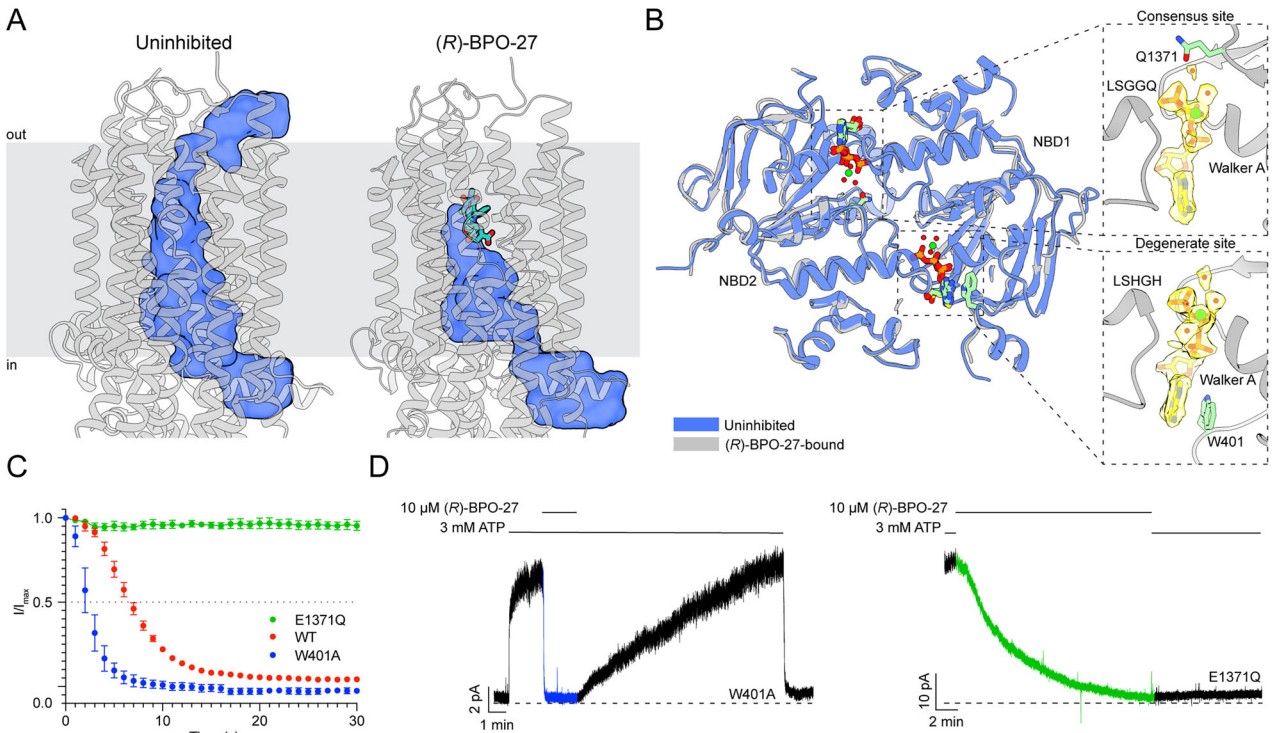

**Fig. 4 | (R)-BPO-27 inhibits by pore blockade and exhibits dimerization-dependent association. A** Comparison of the pore (shown as blue surface) in NBD-dimerized, uninhibited (PDB: 7SVD) and NBD-dimerized, (R)-BPO-27-bound CFTR. The pore is defined as the path accessible to a probe the size of a dehydrated chloride ion (1.7 Å). (R)-BPO-27 is shown as a teal ball-and-stick model. The pore in the uninhibited structure was created from a composite of three probes. The first probe extends from the lateral gate to the selectivity filter. The second probe extends from the selectivity filter just past the extracellular exit. The third probe extends from the extracellular exit into the extracellular vestibule. **B** (*Left*) Overlay of NBDs from uninhibited (PDB: 7SVD; blue) and (R)-BPO-27-bound (gray) CFTR. ATP is shown as green sticks in the uninhibited model and yellow sticks in the (R)-BPO-27-bound model. Active site water molecules are shown as red spheres, and active site magnesium molecules are shown as green spheres. (*Right, Top*) Consensus ATP-binding site in the (R)-BPO-27-bound model. (*Right, Bottom*)

Degenerate ATP-binding site in the (R)-BPO-27-bound model. In both panels, ATP is shown as yellow sticks surrounded by cryo-EM density from the CFTR/(R)-BPO-27 map. **C** Normalized plots of current following 10 μM (R)-BPO-27 addition for CFTR (E1371Q), WT CFTR, and CFTR (W401A). Current values were normalized to the current prior to inhibitor addition. The dotted line indicates 50% maximal current. For each replicate, $t_{1/2}$ values were determined based on the time at which half-maximal current was achieved. Mean $t_{1/2}$ and SE were then calculated across replicates. Data represent means and SE for $n = 3$ (E1371Q) or $n = 4$ (W401A and WT) measurements. The full CFTR (E1371Q) plot is not shown due to its long time course. **D** Representative macroscopic current recording from inside-out membrane patches excised from cells expressing CFTR (W401A) or CFTR (E1371Q). The current following addition of 10 μM (R)-BPO-27 but before inhibitor withdrawal, is shown in blue (W401A) or green (E1371Q).

than the change in $K_m$ we observed (Fig. 1). Second, competitive inhibition requires the apparent $K_m$ to increase linearly with inhibitor concentration, but the $K_m$ varies only slightly as (R)-BPO-27 is raised from 0.5 μM to 5 μM (Fig. 1B). Thus, the behavior of (R)-BPO-27 is inconsistent with true competition.

Rather, our observations are consistent with K-type allostery, in which a ligand bound distantly from the active site modifies substrate affinity[55]. Similar mechanisms have been described for cGMP binding to the catabolite activator protein (CAP)[56] and biaryl small-molecule inhibitors of kinesin spindle protein (KSP)[57]. In the case of CAP, it was shown that interaction with cGMP reduces occupancy of states capable of binding DNA. Similarly, biaryl inhibitor binding was proposed to perturb rearrangements of KSP necessary for ATP binding following hydrolysis[57]. We propose that (R)-BPO-27 binding may affect CFTR via a similar mechanism by stabilizing a conformation with reduced affinity for ATP. However, once bound to ATP, hydrolysis can proceed unimpeded, thus resulting in an increased apparent $K_m$ without affecting the turnover rate. Although our cryo-EM map does not reveal obvious rearrangements within the NBDs that would explain the lower affinity, this is likely because the E1371Q background stabilizes the canonical dimer and masks subtle allosteric distortions[51].

Understanding the mechanisms of CFTR inhibition has important clinical implications. In addition to (R)-BPO-27, the binding site of CFTR$_{inh}$-172 has also been identified[37,38]. Although these two inhibitors bind to a similar region in CFTR, they function through distinct mechanisms (Fig. S5). CFTR$_{inh}$-172 inhibits current not by directly blocking the pore, but by stabilizing a conformation of CFTR in which the selectivity filter is collapsed[37,38]. Additionally, CFTR$_{inh}$-172 inhibits ATP hydrolysis via an allosteric noncompetitive mechanism that alters $k_{cat}$ but not $K_m$[37]. We previously hypothesized that CFTR$_{inh}$-172 does this via a TM 8-mediated mechanism, as this region of CFTR has been shown to modulate both channel and ATPase activities, and CFTR$_{inh}$-172 binding stabilizes a unique conformation of this helix[28,37,38]. Consistent with this hypothesis, (R)-BPO-27 does not affect the conformation of any pore-lining helix and does not perturb ATP turnover. Taken together, these findings highlight the diversity of CFTR inhibition strategies and provide a foundation for developing therapeutics for secretory diarrhea and ADPKD.

## Methods

### Cell culture

Sf9 cells (Gibco, catalog number 11496015, lot number 1670337) were grown at 27 °C in Sf-900 II SFM medium (Gibco) supplemented with 5% (v/v) fetal bovine serum (FBS) and 1% (v/v) antibiotic-antimycotic (Gibco). HEK293S GnTI⁻ cells (American Type Culture Collection (ATCC) CRL-3022, lot number 62430067) were cultured at 37 °C in Freestyle 293 medium (Gibco) supplemented with 2% (v/v) FBS and 1%

(v/v) antibiotic-antimycotic. CHO (Chinese hamster ovary)-K1 cells (ATCC CCL-61, lot number 70014310) were cultured at 37 °C in Dulbecco's Modified Eagle Medium/Nutrient Mixture F-12 (DMEM/F-12) (ATCC) supplemented with 10% (v/v) FBS and 1% (v/v) GlutaMAX (Gibco).

## Patch-clamp electrophysiology

CHO-K1 cells were seeded in 35-mm cell culture dishes (Falcon) twenty-four hours before transfection. Prior to transfection, the media was exchanged to serum-free DMEM/F-12. Cells were transiently transfected with BacMam vector encoding C-terminally green fluorescent protein (GFP)-fused CFTR using Lipofectamine 3000 (Invitrogen). Twelve hours after transfection, the medium was exchanged for DMEM/F-12 supplemented with 2% (v/v) FBS and 1% (v/v) GlutaMAX, and the incubation temperature was reduced to 30 °C. Patch-clamp recordings were carried out after an additional 24 h.

The bath solution was 145 mM NaCl, 2 mM MgCl$_2$, 5 mM KCl, 1 mM CaCl$_2$, 5 mM glucose, 5 mM (4-(2-hydroxyethyl)-1-piperazineethanesulfonic acid (HEPES)) (pH 7.4 with NaOH), and 20 mM sucrose. Pipette solution was 140 mM NMDG, 5 mM CaCl$_2$, 2 mM MgCl$_2$, and 10 mM HEPES (pH 7.4 with HCl). Perfusion solution was 150 mM NMDG, 2 mM MgCl$_2$, 10 mM (ethylene glycol-bis(β-aminoethyl ether)-N,N,N′,N′-tetraacetic acid (EGTA)), 10 mM HEPES (pH 7.4 with NaOH), and 8 mM Tris (pH 7.4 with HCl).

Recordings were carried out using the inside-out patch configuration with local perfusion at the patch. Recording pipettes were pulled from borosilicate glass (outer diameter 1.5 mm, inner diameter 0.86 mm, Sutter) to 1.5–3.0 MΩ resistance. Currents were recorded at 28 °C using an Axopatch 200B amplifier, a Digidata 1550 digitizer, and the pClamp software suite (Molecular Devices). Membrane potential was clamped at −30 mV. Current traces reflect inward currents with inverted signatures. Recordings were low-pass filtered at 1 kHz and digitized at 20 kHz.

For all measurements, CFTR was activated by exposure to PKA (Sigma–Aldrich) and 3 mM ATP. Displayed recordings were low-pass filtered at 100 Hz. Data were analyzed using Clampfit and GraphPad Prism. Values for $t_{1/2}$ were determined either by fitting a one-phase exponential (when the data conformed well to a monoexponential function) or by interpolating the time at which current achieved 50% of the maximum (when fitting was not appropriate) for each replicate and averaging these values. The method used for each analysis is indicated in the corresponding figure legend.

## Protein expression and purification

CFTR constructs were expressed and purified as previously described[58,59]. Bacmids encoding human CFTR fused to a C-terminal PreScission Protease-cleavable GFP tag were generated in *Escherichia coli* DH10Bac cells (Invitrogen). Recombinant baculovirus was produced and amplified in Sf9 cells. HEK293S GnTI⁻ suspension cells, at a density of 2.0–3.0 × 10⁶ cells/mL, were infected with 10% (v/v) P4 baculovirus. Protein expression was induced by the addition of 10 mM (final concentration) sodium butyrate 12 h after infection. The cells were cultured at 30 °C for an additional 48 h and then harvested by centrifugation.

Protein samples for cryo-EM were purified as follows. Cells were solubilized for 75 min at 4 °C in extraction buffer containing 1–1.25% (w/v) 2,2-didecylpropane-1,3-bis-b-D-maltopyranoside, 0.25% (w/v) cholesteryl hemisuccinate, 200 mM NaCl, 20 mM HEPES (pH 7.5 with NaOH), 2 mM MgCl$_2$, 2 mM dithiothreitol (DTT), 20% (v/v) glycerol, 2 mM ATP, 1 µg mL⁻¹ pepstatin A, 1 µg mL⁻¹ aprotinin, 100 µg mL⁻¹ soy trypsin inhibitor, 1 mM benzamidine, 1 mM phenylmethylsulfonyl fluoride, and 3 µg mL⁻¹ DNase I. Lysate was clarified by centrifugation at 75,000 × g for 45 min at 4 °C and mixed with NHS-activated Sepharose 4 Fast Flow resin (GE Healthcare) conjugated with GFP nanobody, which had been pre-equilibrated in extraction buffer. After 2 h, the

resin was packed into a chromatography column and washed with buffer containing 0.06% (w/v) digitonin, 200 mM NaCl, 20 mM HEPES (pH 7.5 with NaOH), 2 mM ATP, and 2 mM MgCl$_2$. The resin was then incubated for 2 h at 4 °C with 0.35 mg mL⁻¹ PreScission Protease to cleave off the GFP tag. The PreScission Protease was removed by dripping eluate through Glutathione Sepharose 4B resin (Cytiva). The protein was then concentrated to yield ~1 mL of sample, which was then phosphorylated by 300 nM PKA for 1 h at room temperature. Finally, protein samples were purified by size-exclusion chromatography at 4 °C using a Superose 6 10/300 GL column (GE Healthcare), equilibrated with a buffer containing 0.03% (w/v) digitonin, 200 mM NaCl, 20 mM HEPES (pH 7.5 with NaOH), 2 mM ATP, and 2 mM MgCl$_2$. Peak fractions were pooled and concentrated. Samples for ATP hydrolysis assays were purified using the same protocol, but in buffers containing KCl, rather than NaCl.

## EM data acquisition and processing

Immediately following size-exclusion chromatography, the CFTR (E1371Q) sample was concentrated to 5 mg/mL (32 µM) and incubated with 8 mM ATP, 10 mM MgCl$_2$, and 92 µM (R)-BPO-27 on ice for approximately 2 h prior to vitrification. 3 mM fluorinated Fos-choline-8 was added to the samples directly before application onto glow-discharged Quantifoil R0.6/1 300 mesh Cu grids. Samples were then vitrified using a Vitrobot Mark IV (Field Electron and Ion Company, FEI).

Cryo-EM images were collected in super-resolution mode on a 300 kV Titan Krios (FEI) equipped with a K3 Summit detector (Gatan) using SerialEM (Table S1). Images were corrected for gain reference and binned by 2. Drift correction was performed using MotionCor2[60]. Contrast transfer function (CTF) estimation was performed using CTFFIND4[61]. All subsequent steps of map reconstruction and resolution estimation were carried out using RELION 3.1 (Table S1; Fig. S1)[62].

## Model building and refinement

Initial protein models were built by fitting the published structure of the NBD-dimerized CFTR (E1371Q) (PDB: 6MSM) into the cryo-EM map using Coot[63]. The model was then adjusted based on the cryo-EM density. (R)-BPO-27 was built into the density and refined in PHENIX[64] using restraints generated by the Global Phasing web server (grade.globalphasing.org). MolProbity[65] was used for geometry validation.

## ATP hydrolysis measurements

Steady-state ATP hydrolysis was measured using an NADH-coupled assay[66]. The assay buffer contained 50 mM HEPES (pH 8.0 with KOH), 150 mM KCl, 2 mM MgCl$_2$, 2 mM DTT, 0.06% (w/v) digitonin, 60 µg mL⁻¹ pyruvate kinase (Roche), 32 µg mL⁻¹ lactate dehydrogenase (Roche), 9 mM phosphoenolpyruvate, 150 µM NADH, 200 nM CFTR, and specified compound concentrations. Aliquots of 27 µL were distributed into a Corning 384-well Black/Clear Flat Bottom Polystyrene NBS Microplate. The reactions were initiated by the addition of ATP. The rate of fluorescence depletion was monitored at $\lambda_{ex} = 340$ nm and $\lambda_{em} = 445$ nm at 28 °C with an Infinite M1000 microplate reader (Tecan). ATP turnover rate was then determined with an NADH standard curve.

## Reporting summary

Further information on research design is available in the Nature Portfolio Reporting Summary linked to this article.

## Data availability

The cryo-EM map of CFTR (E1371Q) in complex with (R)-BPO-27 has been deposited in the Electron Microscopy Data Bank under the accession code EMD-48717[67]. The atomic coordinates have been deposited in the Protein Data Bank (PDB) under accession code 9MXL

(Cryo-EM structure of phosphorylated, ATP-bound human CFTR in complex with (R)-BPO-27)[68]. All other data are available in the main text, supplementary information, or source data for Figs. 1B, C, E, 4, and S3B. The data that support this study are available from the corresponding authors upon request. The following atomic models have also been referenced in this work: 7SVD (Cryo-EM structure of human CFTR in complex with Lumacaftor), 6MSM (Cryo-EM structure of phosphorylated, ATP-bound human CFTR), 5UAK (Cryo-EM structure of dephosphorylated, ATP-free human CFTR), 8UBR (Cryo-EM structure of human CFTR in complex with CFTR$_{inh}$-172). Source data are provided with this paper.

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

## Acknowledgements
We thank Mark Ebrahim, Johanna Sotiris, and Honkit Ng at the Rockefeller University Evelyn Gruss Lipper Cryo-EM Resource Center for their support with collecting electron microscopy data. We thank Jesper Levring for helpful discussions. This work was supported by HHMI to J.C., the Cystic Fibrosis Foundation Therapeutics to K.F., and the National Institutes of Health to P.Y. (Training grant, T32GM007739).

## Author contributions
P.Y. and K.F. determined the cryo-EM structure; P.Y. performed ATPase assays and patch-clamp experiments. P.Y. and J.C. prepared the manuscript.

## Competing interests
The authors declare no competing interests.
