## [Transparent Peer Review file · Nature Communications]

Structure of CFTR bound to (R)-BPO-27 unveils a pore-blockage mechanism

Corresponding Author: Dr Jue Chen

Version 0:

Reviewer comments:

Reviewer #1

(Remarks to the Author)

This is a manuscript addressing the structural mechanism of a potent CFTR inhibitor (R)-BPO-27 discovered in 2011 by Alan Verkman's laboratory. The authors provided high-resolution, cryo-EM structure of the CFTR/(R)-BPO-27 complex, which shows binding of the drug in the internal vestibule of CFTR's pore with little conformational change. Interestingly, the rate of inhibition upon application of (R)-BPO-27 and the rate of recovery from inhibition upon removal of the drug are state dependent: channels with a stable NBD dimer assume a slower rate for either reaction as if NBD separation allows the drug to enter the pore and subsequent NBD dimerization traps the compound in the pore.

Major points:

1. The authors claim that, unlike CFTRinh-172, binding of (R)-BPO-27 does not induce any conformational changes albeit a rightward shift of the curve for [ATP]-dependent hydrolysis rate. This point can be strengthened by providing results of more systematic structural comparisons between drug-free and drug-bound structures.
2. Regarding the relationship between the effects of (R)-BPO-27 and NBD dimerization states, a more rigorous discussion is also warranted. First, previous studies with smFRET from the same group provided nice kinetic data for the rate of NBD dimerization and the rate of dimer separation for all the constructs used in the current study. The authors may want to use those rates for comparison instead of the probability of dimerization. Second, while quantitative analysis was conducted for the rate of inhibition (Figure 4D), a similar analysis for the recovery phase should be done especially for the W401A mutant. Third, I am sure by comparing NBD dimer lifetime (from smFRET) and the recovery rate for W401A-CFTR, the authors will find a large difference, which can then be used to assess the relative role of NBD separation and the strength of drug binding in determining the rate of recovery. Fourth, since the structure of CFTR with separated NBDs is available, it would be nice to look into the binding site of (R)-BPO-27 in that structure again to gauge the role of NBD dimerization in deciding the lifetime of the blocked state. Moreover, by examining the on and off rates of (R)-BPO-27 with CFTR variants with defective NBD dimerization (e.g., G551D or deltaNBD2), the authors can make much stronger conclusions regarding the role of NBD dimerization in determining the effects of (R)-BPO-27.
3. Strictly speaking, the decrease of E1371Q-CFTR currents in Figure 4C does not reflect the inhibitory effect of (R)-BPO-27 since the current will drop in the absence of ATP regardless of the presence of (R)-BPO-27. If the authors' idea is correct, the relaxation rate in this experiment should be similar to the relaxation rate for the same experiment without adding (R)-BPO-27. Furthermore, the failure of ATP to induce any current in the latter part of the experiment is subjected to alternative interpretations. It would be reassuring if ATP is added after removing (R)-BPO-27 for some time and some currents can still be elicited.

Minor points:

1. Page 5, line 7: It seems misleading to conclude that CFTRinh-172 inhibits both CFTR channel activity and ATP hydrolysis to a similar extent. They are not. Close to 100% current inhibition was seen (Figure 1D) by 10 μ M CFTRinh-172, while ATP hydrolysis rate was reduced by ~70% (Figure 1C).
2. Figure S1 needs some more elaborations. For example, what is the purpose of superposition of final model with the 58%-occupancy class (Figure S1B)?
3. Considering the large size of the internal vestibule and previous studies showing the block of CFTR by various large organic anions, I find it interesting that (S)-BPO-27 fails to inhibit CFTR. Have the authors tried it?
4. I also wonder if it is justified to use the ionic radius of chloride (1.7 Å) to define the ion conducting pathway.

Reviewer #2

(Remarks to the Author)

The study by Young et al. reports the high resolution Cryo-EM structures of CFTR bound to (R)-BPO-27, a promising small molecule inhibitor, uncovering the mechanism of CFTR inhibition. Using cryo-EM and electrophysiology, they elucidate the binding site and mechanism of action of (R)-BPO-27, revealing that it inhibits CFTR by directly occluding the chloride-conducting pore rather than competing with ATP binding. This work provides critical insights into the structure-activity relationship of (R)-BPO-27 and offers a molecular basis for optimizing its clinical potential in secretory diarrhea and ADPKD. The study not only clarifies the inhibition mechanism of (R)-BPO-27 but also highlights the diversity of CFTR inhibition strategies.

Major concerns:

1. The study mentions the different inhibition mechanism between (R)-BPO-27 and CFTRinh-172 but lacks a detailed comparison. A more comprehensive analysis of how these two inhibitors interact with CFTR could provide deeper insights into the diversity of CFTR inhibition strategies.
2. While the study proposes that (R)-BPO-27 binding requires NBD separation, the exact mechanism by which the inhibitor accesses its binding site remains unclear. Further investigation into the dynamics of NBD dimerization and separation in the presence of (R)-BPO-27 would strengthen the conclusions.

Reviewer #3

(Remarks to the Author)

The manuscript by Young and coll. reports the identification of the binding site for R-BPO-27, a pharmacological inhibitor of the CFTR chloride channel. In a previous paper by another research team (Kim et al., Mol Pharmacol 2015) this compound was proposed to compete with ATP at nucleotide binding domains (NBDs). Young and coll., using cryo-EM, find surprisingly that R-BPO-27 binds to the CFTR pore, hence quite far from NBDs. Interestingly, R-BPO-27 binding is influenced by NBD dimerization. In fact, it binds when NBDs are separated. Accordingly, the kinetics of inhibition are quite slow on a CFTR mutant (E1371Q) in which NBDs are persistently dimerized.

The study is interesting and provides important information to generate optimized CFTR inhibitors as research tools and possible drugs for human diseases.

Specific comments

In the previous study by Kim et al. (Mol Pharmacol 2015), R-BPO-27 was found to block CFTR currents in a voltage-independent manner. This is surprising given that Young and coll. find that the compound, which is negatively charged, binds to the transmembrane domain of CFTR. I would expect a voltage-dependence of the block. The authors of the present study have done all membrane current recordings at a single voltage (-30 mV). I would suggest to perform experiments at different voltages to verify if CFTR inhibition by this compound is indeed voltage-insensitive.

Introduction, line 17 from top: references 31-36 are cited as papers reporting CFTR inhibitors. However, references 31 and 32 deal instead with CFTR correctors.

Reviewer #4

(Remarks to the Author)

In this manuscript Young and colleagues determined the cryo-EM structure of human CFTR bound to the inhibitor (R)-BPO-27. Previous patch-clamp recordings and molecular docking/simulation studies suggested that (R)-BPO-27 inhibited CFTR channel current by competing with ATP in the NBDs. In contrast, the authors demonstrate here that (R)-BPO-27 binds within the transmembrane helices of CFTR and occludes the central Cl⁻ pore, but still exhibits characteristics of competitive inhibition of ATPase activity in steady-state Michaelis-Menten kinetics. In addition to a high-resolution cryo-EM structure demonstrating binding of (R)-BPO-27 within the central Cl⁻ pore, the authors supplement this structural information by demonstrating that a single point variant (K95A) significantly decreases (R)-BPO-27 inhibition due to loss of a critical electrostatic interaction with the carboxylate moiety on the inhibitor. The authors also demonstrate that (R)-BPO-27 binding is dependent on separation of the NBDs by using mutants previously identified through single-molecule fluorescence measurements to exhibit different probabilities of NBD opening in the presence of nucleotides.

Overall, this structural and functional analysis provides key insight into how (R)-BPO-27 inhibits CFTR channel activity, and will likely provide blueprints for further tailored modifications of the chemical scaffold for improved pharmacological properties. The high resolution cryo-EM analysis was performed by a research group with demonstrated expertise in CFTR structural biology, with all maps, atomic models, and reported statistics appearing to be of high quality. All functional data including ATPase measurements and channel recordings also appear to be high quality, with proper statistical analyses where appropriate. The manuscript is concise and easy to follow, even for a non-expert in CFTR function. While I am supportive of publication of this manuscript, I would also urge the authors to consider the following point for a minor revision...

While this manuscript will undoubtedly provide key insights for readers interested in CFTR and development of inhibitors for this important transporter, there are also important aspects for those interested in ABC transporter and general enzyme mechanisms more broadly. The fact that (R)-BPO-27 displays competitive inhibition characteristics in steady-state ATP hydrolysis experiments but does not directly compete with ATP at the NBDs, runs counterintuitive to classical models of competitive inhibition. While the authors touch briefly on this topic and even suggest that (R)-BPO-27 shifts the K_m for ATP hydrolysis "likely through an allosteric mechanism", expanding the discussion around this phenomenon seems warranted. Incorporating the following points into the manuscript could significantly expand the scope to a broader audience.

1. Only one inhibitor concentration is shown in Figure 1B. While it appears from this data that (R)-BPO-27 exhibits competitive inhibition (increased K_m , same V_{max}), a more formal analysis of enzyme inhibition with expanded inhibitor concentrations would be ideal. I would suggest performing the ATPase assay with a few additional (R)-BPO-27 concentrations to solidify the claim that (R)-BPO-27 displays purely competitive inhibition characteristics, with no decrease in V_{max} even at elevated inhibitor concentrations. Ideally, a Lineweaver-Burk double reciprocal plot could be shown to further solidify this claim.

2. In the discussion, it would be nice to mention other examples of inhibitors that bind at sites far from an enzyme's substrate binding site, but still display competitive inhibition profiles in steady-state kinetics. To this reviewer's knowledge, examples of such "allosteric" inhibitors displaying competitive inhibition kinetic profiles are rather rare. In particular, are there examples of other ABC transporter inhibitors that display similar properties?

3. There is significant discrepancy in how samples were prepared for cryo-EM analysis. In the 3rd paragraph of the discussion, it is stated "the protein sample was incubated with an exceedingly high concentration of (R)-BPO-27 (92 μ M) for several hours prior to vitrification". However, the methods section states that CFTR was "incubated with 8 mM ATP, 10 mM MgCl₂, and 92 μ M (R)-BPO-27 on ice for 30 min". Which of these two statements is correct? If CFTR was indeed incubated with (R)-BPO-27 for several hours prior to vitrification, was this incubation performed on ice or at a different temperature? Was the extended incubation an absolute requirement to obtain the ligand-bound structure?

Version 1:

Reviewer comments:

Reviewer #1

(Remarks to the Author)

No more concerns.

Reviewer #2

(Remarks to the Author)

Reviewer #3

(Remarks to the Author)

No further comments

Reviewer #4

(Remarks to the Author)

The authors have adequately addressed all of my critiques. The additional steady-state ATPase measurements with expanded inhibitor concentrations provide depth to the understanding of the inhibition mechanism, and the discussion of K-type allosteric inhibition is very interesting. I support publication of this manuscript.

Point-to-point response

REVIEWER COMMENTS

Reviewer #1 (Remarks to the Author):

This is a manuscript addressing the structural mechanism of a potent CFTR inhibitor (R)-BPO-27 discovered in 2011 by Alan Verkman's laboratory. The authors provided high-resolution, cryo-EM structure of the CFTR/(R)-BPO-27 complex, which shows binding of the drug in the internal vestibule of CFTR's pore with little conformational change. Interestingly, the rate of inhibition upon application of (R)-BPO-27 and the rate of recovery from inhibition upon removal of the drug are state dependent: channels with a stable NBD dimer assume a slower rate for either reaction as if NBD separation allows the drug to enter the pore and subsequent NBD dimerization traps the compound in the pore.

Major points:

1. The authors claim that, unlike CFTRinh-172, binding of (R)-BPO-27 does not induce any conformational changes albeit a rightward shift of the curve for [ATP]-dependent hydrolysis rate. This point can be strengthened by providing results of more systematic structural comparisons between drug-free and drug-bound structures.

Thank you for the suggestion. We have compared the structures of BPO-bound and drug-free CFTR and calculated the Root Mean Square Deviation (RMSD) between aligned atoms. The results, presented in Supplemental Figure 2, show that, except for a few loop regions, the RMSD value ranges from 0 to 1 Å, indicating a high degree of similarity between the two structures.

Figure R1: Left: Superposition of uninhibited, dimerized CFTR (blue) with the CFTR/(R)-BPO-27 complex (gray). Right: Structure of the CFTR/(R)-BPO-27 complex colored by RMSD from the uninhibited structure.

2.

Regarding the relationship between the effects of (R)-BPO-27 and NBD dimerization states, a more rigorous discussion is also warranted. First, previous studies with smFRET from the same group provided nice kinetic data for the rate of NBD dimerization and the rate of dimer

separation for all the constructs used in the current study. The authors may want to use those rates for comparison instead of the probability of dimerization.

Thank you for this suggestion. However, for technical reasons, the previously published smFRET studies only provided NBD dimerization rate of wtCFTR, not for the E1371Q and W401A variants (Levring *et al.*, 2023). Therefore, we focused on the discussion with dimerization probability, which are known for all three variants.

Second, while quantitative analysis was conducted for the rate of inhibition (Figure 4D), a similar analysis for the recovery phase should be done especially for the W401A mutant. Third, I am sure by comparing NBD dimer lifetime (from smFRET) and the recovery rate for W401A-CFTR, the authors will find a large difference, which can then be used to assess the relative role of NBD separation and the strength of drug binding in determining the rate of recovery.

The recovery rates for WT and W401A are very similar, with $t_{1/2}$ values of 340 ± 51 s and 330 ± 53 s, respectively (Figure R1). As the rate of current recovery is orders of magnitude slower than that of NBD separation, these findings suggest that (R)-BPO-27 binding stabilizes the NBD dimer and traps the channel in an inhibited state. These data are now included as Figure S3 and discussed on page 8.

Figure R2: Left: Normalized plots of current recovery after withdrawal of $10 \mu\text{M}$ (R)-BPO-27 and application of 3 mM ATP for WT CFTR and CFTR (W401A). The dotted line indicates 50% maximal current. Data represent means and SE for 3 measurements. Right: Quantification of recovery $t_{1/2}$ values for WT CFTR and CFTR (W401A). For each replicate, $t_{1/2}$ values were determined based on the time at which half-maximal current was achieved. Mean $t_{1/2}$ and SE were then calculated across replicates. Data represent means and SE for 3 measurements. Significance was tested using a two-tailed Student's t test ($P = 0.84$).

Fourth, since the structure of CFTR with separated NBDs is available, it would be nice to look into the binding site of (R)-BPO-27 in that structure again to gauge the role of NBD dimerization in deciding the lifetime of the blocked state.

Thank you very much for this suggestion, which has led to new insights into (R)-BPO-27 access. Upon analyzing the inhibitor-binding site in the dephosphorylated, NBD-separated CFTR structure (Figure S4), we found that in this conformation, transmembrane helices 8 and 12—both of which undergo local rearrangements upon NBD dimerization—would sterically clash with (R)-BPO-27. This observation suggests that the dephosphorylated, NBD-separated conformation is incompatible with inhibitor binding. To date, only two CFTR conformations have been resolved by cryo-EM; however, smFRET studies indicate the existence of additional

conformational states within the gating cycle. We are now actively pursuing structural studies to capture these intermediates and determine which conformation permits access to (R)-BPO-27. As these investigations are technically challenging and time-consuming, the current manuscript focuses on our primary finding: identifying (R)-BPO-27 as a pore blocker rather than an ATP competitor.

We have revised the text accordingly: “*These observations suggest that (R)-BPO-27 binds to an intermediate sampled during the transition from the NBD-separated state to the fully NBD-dimerized state stabilized by the E1371Q substitution. Elucidating the molecular details of how the inhibitor engages this intermediate remains an active focus of our ongoing research.*”

Moreover, by examining the on and off rates of (R)-BPO-27 with CFTR variants with defective NBD dimerization (e.g., G551D or Δ NBD2), the authors can make much stronger conclusions regarding the role of NBD dimerization in determining the effects of (R)-BPO-27.

As the primary goal of this study is to demonstrate that BPO functions as a pore blocker rather than an ATP competitor, we have limited the rate analysis to the wild-type protein, E1371Q, and W401A variants. While analysis of additional variants could provide further insights, it would be time-consuming and is unlikely to change the central conclusion. Therefore, it falls beyond the scope of this manuscript.

3. Strictly speaking, the decrease of E1371Q-CFTR currents in Figure 4C does not reflect the inhibitory effect of (R)-BPO-27 since the current will drop in the absence of ATP regardless of the presence of (R)-BPO-27. If the authors' idea is correct, the relaxation rate in this experiment should be similar to the relaxation rate for the same experiment without adding (R)-BPO-27. Furthermore, the failure of ATP to induce any current in the latter part of the experiment is subjected to alternative interpretations. It would be reassuring if ATP is added after removing (R)-BPO-27 for some time and some currents can still be elicited.

The time constant of current decay for CFTR (E1371Q) upon ATP removal has been reported as 476 s (Vergani et al., 2005). Since our data do not follow a monoexponential decay, we cannot calculate a time constant in the same manner. Instead, we report the time to half-maximal current decay ($t_{1/2}$), which for the E1371Q variant is 230 s.

We have now performed additional experiments showing that, after BPO removal, the addition of ATP restores the current. A representative recording of these experiments is now presented in Figure S3A.

Minor points:

1. Page 5, line 7: It seems misleading to conclude that CFTR_{inh}-172 inhibits both CFTR channel activity and ATP hydrolysis to a similar extent. They are not. Close to 100% current inhibition was seen (Figure 1D) by 10 μ M CFTR_{inh}-172, while ATP hydrolysis rate was reduced by ~70% (Figure 1C).

We have revised the text as suggested. It now reads: “*This contrasts with CFTR_{inh}-172, which inhibits ATP hydrolysis by 72% and current by 95% at a saturating inhibitor concentration (10 μ M) (Figure 1C-D).*”^{37–39}

2. Figure S1 needs some more elaborations. For example, what is the purpose of superposition of final model with the 58%-occupancy class (Figure S1B)?

We included the superposition to illustrate that the 58% occupancy class adopts the same conformation as the final model, although the particle quality is lower—possibly due to damage during sample preparation. We have clarified this point in the figure legend.

3. Considering the large size of the internal vestibule and previous studies showing the block of CFTR by various large organic anions, I find it interesting that (S)-BPO-27 fails to inhibit CFTR. Have the authors tried it?

This is an interesting point. We have not tried (S)-BPO-27, as it is not commercially available. The Verkman laboratory has showed that at 1 μM concentration, (S)-BPO-27 does not inhibit CFTR (Kim *et al*, 2015).

4. I also wonder if it is justified to use the ionic radius of chloride (1.7 Å) to define the ion conducting pathway.

Although chloride ions at different parts of the pore will have different radii depending on the level of hydration, the smallest relevant radius would be that of a fully dehydrated chloride ion (1.7 Å). For this reason, we used this radius to identify the potential pathway for chloride.

Reviewer #2 (Remarks to the Author):

The study by Young et al. reports the high resolution Cryo-EM structures of CFTR bound to (R)-BPO-27, a promising small molecule inhibitor, uncovering the mechanism of CFTR inhibition. Using cryo-EM and electrophysiology, they elucidate the binding site and mechanism of action of (R)-BPO-27, revealing that it inhibits CFTR by directly occluding the chloride-conducting pore rather than competing with ATP binding. This work provides critical insights into the structure-activity relationship of (R)-BPO-27 and offers a molecular basis for optimizing its clinical potential in secretory diarrhea and ADPKD. The study not only clarifies the inhibition mechanism of (R)-BPO-27 but also highlights the diversity of CFTR inhibition strategies.

Thank you.

Major concerns:

1. The study mentions the different inhibition mechanism between (R)-BPO-27 and CFTR_{inh}-172 but lacks a detailed comparison. A more comprehensive analysis of how these two inhibitors interact with CFTR could provide deeper insights into the diversity of CFTR inhibition strategies.

As suggested, we have now added a figure to compare the binding sites of the two inhibitors (Figure S5) and expanded discussion to compare their mechanisms. On page 9: “*Although these two inhibitors bind to a similar region in CFTR, they function through distinct mechanisms (Figure S5). CFTR_{inh}-172 inhibits current not by directly blocking the pore, but by stabilizing a conformation of CFTR in which the selectivity filter is collapsed.^{37,38} Additionally, CFTR_{inh}-172 inhibits ATP hydrolysis via an allosteric noncompetitive mechanism that alters k_{cat} but not K_m .³⁷ We previously hypothesized that CFTR_{inh}-172 does this via a TM 8-mediated mechanism, as*

this region of CFTR has been shown to modulate both channel and ATPase activities, and CFTR_{inh}-172 binding stabilizes a unique conformation of this helix.^{28,37,38}

Figure R3: (A) Transmembrane regions of CFTR/(R)-BPO-27 and CFTR/CFTR_{inh}-172 complexes. Inhibitors are shown as sticks. Helices stabilized in a unique conformation in the CFTR_{inh}-172-bound structure are shown as blue cylinders in both models. (B) Comparison of the binding sites of (R)-BPO-27 (Left) and CFTR_{inh}-172 (Right). Inhibitors are shown as sticks. Three residues that form part of the binding site are shown.

2. While the study proposes that (R)-BPO-27 binding requires NBD separation, the exact mechanism by which the inhibitor accesses its binding site remains unclear. Further investigation into the dynamics of NBD dimerization and separation in the presence of (R)-BPO-27 would strengthen the conclusions.

We agree. Understanding the exact mechanism by which the inhibitor binds is a topic we are actively pursuing, but we believe it is beyond the scope of the current manuscript. In the revision, we now state that “Given the large size of (R)-BPO-27 relative to the cytoplasmic pore opening (Figure 4), its access to the binding site likely requires separation of the NBDs to open the cytosolic route. The inverse correlation between inhibition rate and the probability of NBD dimerization supports this notion. However, mapping the inhibitor-binding site onto the dephosphorylated, NBD-separated CFTR structure reveals that TM helices 8 and 12, which undergo local rearrangements upon NBD dimerization,^{52,53} would obstruct the (R)-BPO-27 binding site in the NBD-separated state (Figure S4). These observations suggest that (R)-BPO-27 binds to an intermediate sampled during the transition from the NBD-separated state to the fully NBD-dimerized state stabilized by the E1371Q substitution. Elucidating the molecular details of how the inhibitor engages this intermediate remains an active focus of our ongoing research.”

Reviewer #3 (Remarks to the Author):

The manuscript by Young and coll. reports the identification of the binding site for R-BPO-27, a pharmacological inhibitor of the CFTR chloride channel. In a previous paper by another research team (Kim et al., Mol Pharmacol 2015) this compound was proposed to compete with ATP at nucleotide binding domains (NBDs). Young and coll., using cryo-EM, find surprisingly that R-BPO-27 binds to the CFTR pore, hence quite far from NBDs. Interestingly, R-BPO-27 binding is influenced by NBD dimerization. In fact, it binds when NBDs are separated. Accordingly, the kinetics of inhibition are quite slow on a CFTR mutant (E1371Q) in which NBDs are persistently dimerized.

The study is interesting and provides important information to generate optimized CFTR inhibitors as research tools and possible drugs for human diseases.

Thank you.

Specific comments

In the previous study by Kim et al. (Mol Pharmacol 2015), R-BPO-27 was found to block CFTR currents in a voltage-independent manner. This is surprising given that Young and coll. find that the compound, which is negatively charged, binds to the transmembrane domain of CFTR. I would expect a voltage-dependence of the block. The authors of the present study have done all membrane current recordings at a single voltage (-30 mV). I would suggest to perform experiments at different voltages to verify if CFTR inhibition by this compound is indeed voltage-insensitive.

The experiments suggested by the reviewer have been previously carried out by another group. It was reported that (R)-BPO-27 inhibition is independent of voltage (Kim et al., 2015).

Introduction, line 17 from top: references 31-36 are cited as papers reporting CFTR inhibitors. However, references 31 and 32 deal instead with CFTR correctors.

References 31 and 32 described the discovery of GLPG 2737, which functions as a folding correct as well as an inhibitor. The titles of these papers only communicate its corrective efficacy, but the full manuscripts discuss its ability to inhibit CFTR as well.

Reviewer #4 (Remarks to the Author):

In this manuscript Young and colleagues determined the cryo-EM structure of human CFTR bound to the inhibitor (R)-BPO-27. Previous patch-clamp recordings and molecular docking/simulation studies suggested that (R)-BPO-27 inhibited CFTR channel current by competing with ATP in the NBDs. In contrast, the authors demonstrate here that (R)-BPO-27 binds within the transmembrane helices of CFTR and occludes the central Cl⁻ pore, but still exhibits characteristics of competitive inhibition of ATPase activity in steady-state Michaelis-Menten kinetics. In addition to a high-resolution cryo-EM structure demonstrating binding of (R)-BPO-27 within the central Cl⁻ pore, the authors supplement this structural information by demonstrating that a single point variant (K95A) significantly decreases (R)-BPO-27 inhibition due to loss of a critical electrostatic interaction with the carboxylate moiety on the inhibitor. The authors also demonstrate that (R)-BPO-27 binding is dependent on separation of the NBDs by using mutants previously identified through single-molecule fluorescence measurements to exhibit different probabilities of NBD opening in the presence of nucleotides.

Overall, this structural and functional analysis provides key insight into how (R)-BPO-27 inhibits CFTR channel activity, and will likely provide blueprints for further tailored modifications of the chemical scaffold for improved pharmacological properties. The high resolution cryo-EM analysis was performed by a research group with demonstrated expertise in CFTR structural biology, with all maps, atomic models, and reported statistics appearing to be of high quality. All functional data including ATPase measurements and channel recordings also appear to be high quality, with proper statistical analyses where appropriate. The manuscript is concise and easy to follow, even for a non-expert in CFTR function. While I am supportive of publication of this manuscript, I would also urge the authors to consider the following point for a minor revision...

While this manuscript will undoubtedly provide key insights for readers interested in CFTR and development of inhibitors for this important transporter, there are also important aspects for those interested in ABC transporter and general enzyme mechanisms more broadly. The fact that (R)-BPO-27 displays competitive inhibition characteristics in steady-state ATP hydrolysis experiments but does not directly compete with ATP at the NBDs, runs counterintuitive to classical models of competitive inhibition. While the authors touch briefly on this topic and even suggest that (R)-BPO-27 shifts the K_m for ATP hydrolysis “likely through an allosteric mechanism”, expanding the discussion around this phenomenon seems warranted. Incorporating the following points into the manuscript could significantly expand the scope to a broader audience.

1. Only one inhibitor concentration is shown in Figure 1B. While it appears from this data that (R)-BPO-27 exhibits competitive inhibition (increased K_m , same V_{max}), a more formal analysis of enzyme inhibition with expanded inhibitor concentrations would be ideal. I would suggest performing the ATPase assay with a few additional (R)-BPO-27 concentrations to solidify the claim that (R)-BPO-27 displays purely competitive inhibition characteristics, with no decrease in V_{max} even at elevated inhibitor concentrations. Ideally, a Lineweaver-Burk double reciprocal plot could be shown to further solidify this claim.

Thank you for this suggestion. We have now performed ATP hydrolysis measurements with three different concentrations of (R)-BPO-27: 0.5, 1, and 5 μM . These results, presented in Figure 1B-C, demonstrate that the k_{cat} and K_m are similar at all three concentrations, consistent with the conclusion that (R)-BPO-27 is not a competitive inhibitor. We have also added a section to the discussion considering these findings in light of the expected K_m change if the inhibitor were truly competing with ATP. By our calculations, based on the IC_{50} for (R)-BPO-27 reported in Kim *et al.*, 2015 (4 nM), 1 μM of (R)-BPO-27 should increase the K_m of ATP hydrolysis to nearly 900 mM if it is a true competitor.

We have now revised the manuscript to discuss these points and expanded Figure 1B-C. On page 8: “While the structure shows that (R)-BPO-27 binds deep in the pore, distant from the ATP-binding sites, functional assays reveal modest right-shifts in both the EC_{50} for ATP-gated current⁴⁵ and the apparent K_m for ATP hydrolysis (Figure 1B). Although a change in K_m is often evidence for competitive inhibition, two observations contradict that interpretation here. First, if (R)-BPO-27 were a true competitive inhibitor, its IC_{50} of 0.5 nM, converted to a K_i with the Cheng-Prusoff equation,⁵⁴ predicts that 1 μM inhibitor should push the apparent K_m into the molar range, which is orders of magnitude greater than the change in K_m we observed (Figure 1). Second, competitive inhibition requires the apparent K_m to increase linearly with inhibitor concentration, but the K_m varies only slightly as (R)-BPO-27 is raised from 0.5 μM to 5 μM (Figure 1B).”

Figure R4: (B) Effect of (R)-BPO-27 on steady-state ATP hydrolysis by PKA-phosphorylated WT CFTR. Data represent means and SE from 3 (0.5 μM, 1 μM, and 5 μM (R)-BPO-27) or 5 (untreated) measurements and are fitted with the Michaelis-Menten equation. Without (R)-BPO-27, the K_m was 0.36 ± 0.11 mM; with 0.5 μM (R)-BPO-27, the K_m was 1.4 ± 0.6 mM; with 1 μM (R)-BPO-27, the K_m was 1.6 ± 0.4 mM; with 5 μM (R)-BPO-27, the K_m was 1.8 ± 0.8 mM. The k_{cat} values are plotted in panel C. (C) Maximal turnover rates of WT CFTR in the absence and presence of inhibitors. The k_{cat} values based on fitted Michaelis-Menten curves: in the absence of inhibitors: 21 ± 3 ATP/protein/minute; in the presence of 0.5 μM (R)-BPO-27: 25 ± 2 ATP/protein/minute; in the presence of 1 μM (R)-BPO-27: 27 ± 2 ATP/protein/minute; in the presence of 5 μM (R)-BPO-27: 26 ± 2 ATP/protein/minute; in the presence of 10 μM CFTR_{inh}-172: 5.8 ± 1.1 ATP/protein/minute. Data represent means and SE from 3 (with all concentrations of (R)-BPO-27), 4 (with CFTR_{inh}-172), or 5 (untreated) titrations. Statistical significance was tested by one-way ANOVA (** $P = 10^{-3}$).

2. In the discussion, it would be nice to mention other examples of inhibitors that bind at sites far from an enzyme's substrate binding site, but still display competitive inhibition profiles in steady-state kinetics. To this reviewer's knowledge, examples of such "allosteric" inhibitors displaying competitive inhibition kinetic profiles are rather rare. In particular, are there examples of other ABC transporter inhibitors that display similar properties?

This type of allostery, termed as K-type system, refers to binding of an allosteric ligand changes the affinity of the substrate and is indeed rare. To the best of our knowledge, this is the first example in the ABC transporter family. We have revised the manuscript to discuss this point and referenced relevant papers. On page 8: "Rather, our observations are consistent with K-type allostery, in which a ligand bound distantly from the active site modifies substrate affinity.⁵⁵ Similar mechanisms have been described for cGMP binding to the catabolite activator protein (CAP)⁵⁶ and biaryl small molecule inhibitors of kinesin spindle protein (KSP).⁵⁷ In the case of CAP, it was shown that interaction with cGMP reduces occupancy of states capable of binding DNA. Similarly, biaryl inhibitor binding was proposed to perturb rearrangements of KSP necessary for ATP binding following hydrolysis.⁵⁷ We propose that (R)-BPO-27 binding may affect CFTR via a similar mechanism by stabilizing a conformation with reduced affinity for ATP. However, once bound to ATP, hydrolysis can proceed unimpeded, thus resulting in an increased apparent K_m without affecting the turnover rate. Although our cryo-EM map does not reveal obvious rearrangements within the NBDs that would explain the lower affinity, this is likely because the E1371Q background stabilizes the canonical dimer and masks subtle allosteric distortions.⁵¹"

3. There is significant discrepancy in how samples were prepared for cryo-EM analysis. In the

3rd paragraph of the discussion, it is stated “the protein sample was incubated with an exceedingly high concentration of (R)-BPO-27 (92 μ M) for several hours prior to vitrification”. However, the methods section states that CFTR was “incubated with 8 mM ATP, 10 mM MgCl₂, and 92 μ M (R)-BPO-27 on ice for 30 min”. Which of these two statements is correct? If CFTR was indeed incubated with (R)-BPO-27 for several hours prior to vitrification, was this incubation performed on ice or at a different temperature? Was the extended incubation an absolute requirement to obtain the ligand-bound structure?

Thank you for pointing this out. We have corrected the methods section. It now states that: *“...the CFTR (E1371Q) sample was concentrated to 5 mg/mL (32 μ M) and incubated with 8 mM ATP, 10 mM MgCl₂, and 92 μ M (R)-BPO-27 on ice for approximately 2 hours prior to vitrification.”*

We did not test if the extended incubation is an absolute requirement, which would require collecting many datasets and determining the structures at different time points. For this reason, we have now revised the manuscript to remove the discussion of incubation time.